# Kynurenines as a Novel Target for the Treatment of Inflammatory Disorders

**DOI:** 10.3390/cells13151259

**Published:** 2024-07-26

**Authors:** Adrian Mor, Anna Tankiewicz-Kwedlo, Marianna Ciwun, Janina Lewkowicz, Dariusz Pawlak

**Affiliations:** 1Department of Pharmacodynamics, Medical University of Bialystok, A. Mickiewicza 2C, 15-222 Bialystok, Poland; adrian.mor@outlook.com (A.M.); marianna.ciwun@gmail.com (M.C.); dariusz.pawlak@umb.edu.pl (D.P.); 2Department of Internal Medicine and Metabolic Diseases, Medical University of Bialystok, M. Sklodowskiej-Curie 24A, 15-276 Bialystok, Poland; janka.lewkowicz@gmail.com

**Keywords:** tryptophan, kynurenine pathway, kynurenine, indoleamine 2,3-dioxygenase, tryptophan 2,3-dioxygenase, kynurenine 3-monooxygenase, immunosuppression, inflammatory disorders

## Abstract

This review discusses the potential of targeting the kynurenine pathway (KP) in the treatment of inflammatory diseases. The KP, responsible for the catabolism of the amino acid tryptophan (TRP), produces metabolites that regulate various physiological processes, including inflammation, cell cycle, and neurotransmission. These metabolites, although necessary to maintain immune balance, may accumulate excessively during inflammation, leading to systemic disorders. Key KP enzymes such as indoleamine 2,3-dioxygenase 1 (IDO1), indoleamine 2,3-dioxygenase 2 (IDO2), tryptophan 2,3-dioxygenase (TDO), and kynurenine 3-monooxygenase (KMO) have been considered promising therapeutic targets. It was highlighted that both inhibition and activation of these enzymes may be beneficial, depending on the specific inflammatory disorder. Several inflammatory conditions, including autoimmune diseases, for which modulation of KP activity holds therapeutic promise, have been described in detail. Preclinical studies suggest that this modulation may be an effective treatment strategy for diseases for which treatment options are currently limited. Taken together, this review highlights the importance of further research on the clinical application of KP enzyme modulation in the development of new therapeutic strategies for inflammatory diseases.

## 1. Introduction

Tryptophan (TRP) is an essential biogenic amino acid characterized by its extensive metabolism through several metabolic pathways, with the kynurenine pathway (KP) being mostly responsible for its catabolism [1]. Metabolites of KP, due to their biological activity, participate in the regulation of numerous physiological processes, including cell cycle [2], inflammation [3], and neurotransmission [4], as well as being involved in the development of pathological conditions [5,6,7,8]. Within this pathway, TRP is oxidized to N-formylkynurenine by tryptophan 2,3-dioxygenase (TDO) and two indoleamine 2,3-dioxygenase isoforms (IDO-1 and IDO-2). Under physiological conditions, they catalyze the same reaction in parallel but have different tissue distributions [9]. IDO1 is observed in almost all body tissues, while TDO activity is highest in the liver [10]. In the next step, N-formylkynurenine is converted to kynurenine (KYN) by formamidase. The presence of KYN has been confirmed in blood, most peripheral tissues of the body, and the brain. KYN is metabolized by the three branches of KP, resulting in the formation of kynurenic acid (KYNA), anthranilic acid (AA), and 3-hydroxykynurenine (3-HKYN). The conversion of KYN to 3-HKYN is catabolized by kynurenine 3-monooxygenase (KMO), while KYNA is by kynurenine aminotransferase (KAT). In turn, 3-HKYN is converted to xanthurenic acid (XA) by the KAT and to 3-HAA after modification with the kynureninase (KYNU). The presence of KMO, KAT, and KYNU has been confirmed in almost all body tissues [11]. 3-HAA is metabolized by 3-hydroxyanthranilic acid 3,4-dioxygenase (3-HAO) to aminocarboxymuconatesemialdehyde (ACMS), which is converted by ACMS decarboxylase to aminomuconicsemialdehyde. This compound undergoes non-enzymatic cyclization to picolinic acid (PA) or is non-enzymatically transformed to quinolinic acid (QA), which is converted by quinolinate phosphoribosyltransferase into an oxidized form of nicotinamide adenine dinucleotide, a coenzyme directly participating in numerous cellular metabolic processes [4]. There is strong evidence demonstrating that in the changes in the activity of KP, enzymes are observed in the development of a wide range of systemic disorders [11], including autoimmune [12], infectious [13], and metabolic diseases [14]. Regulation of the enzyme activity of the kynurenine pathway (KP) involves complex genetic, epigenetic, and environmental mechanisms, both in physiological and pathological conditions. Genetic factors appear to play a key role in controlling the activity of the KP enzyme. For example, specific polymorphisms of the IDO1 and KMO genes may affect their expression level and functional activity, and thus influence the production of both neuroprotective and neurotoxic metabolites. Genetic mutations in these enzymes have been associated with a variety of diseases, including neurodegenerative and psychiatric disorders [15,16]. Epigenetic modifications such as DNA methylation and histone acetylation also regulate the expression of KP enzymes. Environmental factors and disease states may influence these modifications. For example, inflammation can induce IDO1 expression by activating transcription factors such as NF-κB and STAT1, which bind to the promoter regions of the IDO1 gene. Moreover, epigenetic drugs that modify histone acetylation patterns can alter the expression of KP enzymes, thereby affecting pathway activity [4,17]. Environmental factors such as stress, infection, and dietary ingredients significantly influence the activity of the KP enzyme. Inflammatory cytokines such as IFN-γ and TNF can increase IDO1 and TDO levels, increasing the conversion of TRP to KYN [18]. In pathological conditions such as cancer or chronic infection, persistent activation of these enzymes can lead to immunosuppression and contribute to disease progression. End-product inhibition is another regulatory mechanism in which accumulated downstream metabolites, such as QA, can inhibit upstream enzymes, maintaining metabolic balance [19]. Research on the biological role of the KP has led authors to conclude that targeting enzymes within this pathway may be an effective method of treating inflammatory diseases, including those with limited therapeutic options or currently considered incurable. This literature review aims to identify these enzymes as potential therapeutic targets.

## 2. The Role of the Kynurenine Pathway in the Immune Response

The KP plays an important role in maintaining the balance of the immune system, as indicated by the close associations between KP activation and the expression levels of pro-inflammatory cytokines [20]. The activation of KP is induced mainly by pro-inflammatory factors such as interleukins (IL-1 and IL-6), tumor necrosis factor (TNF), and interferon-γ (IFN-γ), and it is observed during inflammation [21]. Numerous studies have demonstrated that KP activation is responsible for the negative feedback suppression loop of immune activation [12,22,23,24]. It enhances immune tolerance by increasing the concentration of KYN and its downstream metabolites. KYN is a compound with potent immunosuppressive properties and is an agonist of the aryl hydrocarbon receptor (AhR), which plays an important role in regulating numerous cellular signaling pathways and maintaining cellular homeostasis. Another AhR agonist is KYNA [25]. The AhR is a ligand-activated transcription factor that is widely distributed in various tissues throughout the body. AhR is highly expressed in hepatocytes, where it plays a key role in the metabolism of xenobiotics and endogenous compounds. Additionally, AhR is expressed in a variety of immune cells, including T cells, B cells, dendritic cells, and macrophages, influencing the immune response and inflammation (Figure 1). This effect seems to be the most important among the systemic effects because activating this receptor leads to an intensified immunosuppressive effect. Keratinocytes in the epidermis also express AhR, contributing to skin homeostasis and response to environmental toxins. This receptor is present in lung cells, where it helps respond to pollutants in the air. Intestinal epithelial cells and immune cells in the intestine express AhR, which is involved in maintaining intestinal homeostasis and modulating intestinal microbiota. AhR is expressed in the reproductive organs, where it may influence reproductive development and function [12,18]. Neurons and glial cells in the central nervous system express AhR, indicating a role in neurodevelopment and neuroprotection. These wide distributions highlight the importance of AhRs in regulating many physiological processes and responding to environmental stimuli. AhR activation is closely associated with a nuclear transition that relies on a sequence of positively charged amino acids known as the nuclear localization signal (NLS). The NLS consists of two segments (bipartite NLS) and is located within the conserved basic helix–loop–helix (bHLH) domain in the N-terminal part of the protein. In its unliganded state, AhR predominantly resides in the cytoplasm, bound to a chaperone complex comprising two molecules of HSP90, and single molecules of co-chaperone p23 and hepatitis x-associated protein-2 (XAP2). Most AhR ligands are hydrophobic, including KYN and KYNA, allowing them to cross the cell membrane through simple diffusion [26]. KYN can cross cell membranes and penetrate tissue readily [4]. Additionally, kynurenine uptake in T cells is mediated by the System L Amino Acid Transporter SLC7A5 [27]. When a ligand binds to cytosolic AhR, it induces a conformational change that exposes the NLS, facilitating its recognition by nuclear transporters. Specifically, members of the importin (IMP) superfamily, such as IMPβ1 or its adaptor protein IMPα, can recognize the exposed NLS. Following this recognition, AhR, along with the IMPα/β1 heterodimer, is transported into the nucleus through the nuclear pore complexes (NPCs). On the nuclear side of NPCs, IMPβ1 binds to RanGTP (Ras-related nuclear protein), which leads to the release of the NLS-cargo. This process allows AhR to enter the nucleus and exert its regulatory functions. The combination of kynurenine and AHR leads to the formation of a complex with the nuclear translocator molecule (Arnt) in the nucleus, influencing changes in gene transcription. Activation of AHR by KYN or KYNA induces FoxP3, which promotes the differentiation of naïve CD4+ cells into a Treg cell phenotype and also inhibits RORγt expression, preventing cell maturation into Th17 cells. Thus, KYN and KYNA are at the center of the immune seesaw, which can be pro-inflammatory (via Th17 cells) or anti-inflammatory (via Tregs) [4].

Its activation upregulates the IL-6 and signal transducer and activator of transcription 3 (STAT3) expression, inducing the general control non-repressible-2 kinase and mammalian target of rapamycin kinase pathways [28,29]. Thus, increased synthesis of KYN, mainly by IDO1, suppresses the immune system response, causing inactivation and apoptosis of Th1 and effector T cells, as well as activation of immunosuppressive T regulatory cells (Tregs) [30]. KYN-activated Tregs can upregulate IDO1 expression in dendritic cells (DCs) in response to antigen presentation, further increasing immunosuppression [18] (Figure 1).

This indicates that IDO1 expression can also be elevated by autocrine stimulation in inflammatory conditions [29,30,31]. Conversely, a decrease in IL-6 or STAT3 expression reduces IDO1 activity and KYN synthesis [31,32], while blocking AhR signaling restores T cell proliferation and activation [33]. This suggests that KP metabolites and enzymes help restore immune homeostasis in physiological conditions, preventing excessive immune responses [34]. Additionally, the administration of KYNA in animal models has been shown to exert significant anti-inflammatory effects and lower pro-inflammatory cytokine secretion [35]. However, an excessive level of KYN in the body can lead to overactivation of AhR, accelerating cellular aging processes [36,37,38,39] and increasing their apoptosis rate [28,40,41,42]. This can disturb physiological processes in the tissues of virtually all organs and exacerbate systemic disorders associated with numerous diseases. Increased KYN synthesis also upregulates KMO activity [43,44,45], contributing to the accumulation of 3-HKYN and its metabolites [46]. Due to the pro-oxidative properties of these compounds, their excess may intensify tissue and organ damage, leading to the exacerbation of inflammatory processes and disease symptoms [47,48] (Figure 2).

As mentioned above, an imbalance in the level of pro-inflammatory mediators impacts KP enzyme activity. Therefore, chronic inflammation is usually accompanied by overactivation of this pathway, leading to the accumulation of its metabolites. Due to their significant biological activity, their excess may disturb various biological processes, intensifying systemic homeostasis disorders caused by the ongoing disease. Since pharmacological modulation of KP activity can affect the immune response and inhibit processes involved in the progression of numerous diseases, it seems to be a promising solution for treating various disorders accompanied by pathologically severe inflammation, including autoimmune diseases [11,15,49]. However, it should be noted that the mechanisms responsible for alterations in the activation of signaling pathways associated with changes in KP activity are not fully understood.

## 3. The Involvement of the Kynurenine Pathway in the Pathogenesis of Selected Autoimmune Diseases

### 3.1. Pharmacological Inhibition of Enzyme of IDO1 and TDO Activity

#### 3.1.1. Rheumatoid Arthritis

It is well known that T cells are involved in the development of rheumatoid arthritis (RA), both through direct tissue infiltration and indirectly via the release of pro-inflammatory cytokines [50,51,52]. Furthermore, B cells also participate in the progression of autoimmune reactions in RA [53,54,55]. They synthesize autoantibodies and trigger the autoimmune response via the presentation of autoantigens to T cells and the release of pro-inflammatory cytokines [50,56]. Treatment with anti-CD20 antibodies, i.e., rituximab, depletes the B cell population and attenuates RA symptoms [57,58,59]. However, they rapidly return after the repopulation of B cells into their autoantibody-secreting terminal form. It was demonstrated that the relief of arthritis was not due to a reduction in regulatory T cells or an altered T helper cell phenotype but was due to a reduced autoreactive B cell response [60]. At this point, it is worth noting that IDO1 activity is essential for the differentiation of the autoreactive B cell profile at the initiation of the autoimmune response [18,60]. At least two potential mechanisms have been identified by which the IDO1 pathway may regulate B cell activity. The first mechanism of action of IDO1-mediated suppression involves intracellular and microenvironmental tryptophan consumption combined with activation of the integrated stress response (ISR) kinase, General Control Non-depressant 2 (GCN2) [61]. IDO1-driven activation of GCN2 signaling resulted in proliferation arrest in naïve T cells and apoptosis of inflammatory T cells while promoting the differentiation and activation of FoxP3+ Tregs [61]. Another mechanism of immunity regulation by IDO1 involves the aryl hydrocarbon receptor (AhR), with one of its ligands being kynurenine, for example. However, as demonstrated by Shinde et al., kynurenines do not alter the B cell response to NP-Ficoll or NP-LPS in vitro or in vivo, indicating that GCN2 signals may be the dominant mechanism by which IDO1 regulates T cell-independent immune responses [40,62].

1-methyl-tryptophan (1-MT), a mixture of the two racemic isoforms 1-methyl-D-tryptophan and 1-methyl-L-tryptophan, is an IDO inhibitor used in preclinical research, which mainly inhibits the activity of IDO1 isoform [63,64,65,66]. Its administration to arthritis mice effectively inhibits the aforementioned process [67]. This way, pharmacological inhibition of IDO1 can prevent the recurrence of autoimmune arthritis symptoms caused by the B cell repopulation [60]. Therefore, IDO1 inhibitor seems to be useful in the combination treatment with B cell depletion therapy, as a potentially effective strategy in the management of RA [10,67] (Figure 3).

#### 3.1.2. Morphea and Cutaneous Sclerosing Disorders

The pathogenesis of morphea and other cutaneous sclerosing disorders remains poorly understood [68,69]. They are considered to be autoimmunological diseases; therefore, abnormalities in tryptophan metabolism may be associated with their pathogenesis [70,71]. Their current therapy is directed to the suppression of the autoimmune response [69,72]. In this case, manipulation of the KP activity might also be an additional therapeutic option. A clinical study observed that L-tryptophan supplementation was responsible for an outbreak of eosinophilia syndrome, which caused skin hardening resembling eosinophilic fasciitis [73]. Moreover, several cases of scleroderma-like diseases have been reported during the treatment of parkinsonism with combinations of L-5-hydroxytryptophan and levodopa with benserazide, which is a strong kynureninase inhibitor [74]. Tranilast (N-[3′,4′-dimethoxycinnamoyl]-anthranilic acid) is a 3-HAA derivative capable of targeting this pathway [72]. It acts as a competitive inhibitor of both IDO and TDO [70]. Additionally, it inhibits the activity of transforming growth factor β, which is normally enhanced by 3-HAA [75]. Tranilast inhibits the expression of the transcriptional coactivator cAMP response element binding protein (CBP) by disrupting the interaction between NF-κB and CBP. At therapeutically relevant concentrations (50 µg/mL), tranilast has been proven to inhibit NF-κB-dependent transcriptional activation by interfering with NF-κB/CBP association. It probably contributes to the anti-inflammatory effect of tranilast by inhibiting the transcription of the ICAM-1-κB and E-selectin-κB reporter genes. Furthermore, tranilast significantly inhibited interleukin 6 secretion in human umbilical vein endothelial cells [76]. For all these reasons, it could be useful in the treatment of the above-mentioned diseases. Several clinical studies confirm the role of tranilast in the treatment of scleroderma, primarily as an adjunct to established therapies. It has been proven that only combined therapy with systemic drugs and the combination of local betamethasone and tranilast showed statistically significant improvement over 3 months in variants of this disease with a worse prognosis, i.e., periarticular variants, frontoparietal variants, and generalized variants (Figure 3) [77].

#### 3.1.3. Systemic Lupus Erythematosus

Fatigue and depression belong to the most disturbing symptoms observed in the majority of patients with systemic lupus erythematosus (SLE) [78]. They affect even individuals with a mild form of the disease. Research has shown that the kynurenine pathway (KP) is overactivated in the course of SLE, resulting from the increased levels of pro-inflammatory cytokines commonly observed during this disease [79,80]. The increase in IDO activity seems to be associated with the development of some symptoms accompanying the SLE [78]. The severity of SLE-related fatigue positively correlates with serum KYN and QA levels [81,82]. Research has shown that the kynurenine pathway (KP) is overactivated in the course of SLE, resulting from the increased levels of pro-inflammatory cytokines commonly observed during this disease (Figure 3) [78,83].

### 3.2. Pharmacological Stimulation of IDO1 Activity in Autoimmune Diabetes

#### Autoimmune Diabetes

Mondanelli et al. have illustrated that during inflammatory states, IDO1 undergoes proteasomal degradation by dendritic cells (DCs), shifting their function from immunoregulatory to immunostimulatory [84]. Bortezomib (BTZ), a proteasome inhibitor approved for treating multiple myeloma [85], also indirectly modulates immune cell activation [86]. While BTZ downregulates IFN-γ-induced IDO expression in nasopharyngeal carcinoma cells, it restores IDO1 protein levels in DCs from non-obese diabetic mice by inhibiting proteasomal degradation. This DC-mediated mechanism contributes to immune response suppression [87]. These properties of BTZ hold promise for treating autoimmune diseases [85,88]. In vivo administration of BTZ prevents autoimmune diabetes development through IDO1- and DC-dependent mechanisms. While it exhibits limited therapeutic efficacy in monotherapy, combining suboptimal doses with an autoimmune-preventive anti-CD3 antibody triggers disease reversal in diabetic mice [84]. The therapeutic outcome parallels full-dose anti-CD3 treatment but with reduced adverse effect severity. Hence, these findings suggest BTZ’s potential in managing autoimmune diabetes and underscore the role of IDO1-mediated immune regulation in its progression [89]. Additionally, BTZ has shown the potential to restore immune balance in autoimmune contexts by bolstering the IDO1-dependent mechanism [84,90].

## 4. Non-Autoimmune Inflammatory Disorders

### 4.1. Pharmacological Modulation Activity of IDO1 Activity

#### 4.1.1. Inhibition of the Enzyme

##### Metabolic Diseases

Both preclinical and clinical studies have established the influence of the kynurenine pathway (KP) on the severity of metabolic diseases [91,92,93,94]. KP metabolites exert signaling effects across various tissues, linking diverse physiological and pathological processes [22,95]. Targeting the KP to concurrently modulate inflammation and energy metabolism holds significant promise for expanding preventive and therapeutic strategies for cardiologic and metabolic disorders, among others [14,96,97,98]. Enhanced comprehension of the mechanisms governing KP enzyme expression or activity is crucial, as they dictate the balance between metabolites. Manipulating the KP through enzyme inhibitors or metabolites presents a novel therapeutic avenue for addressing atherosclerosis [99,100,101], obesity [102,103,104], glucose intolerance [96], impaired insulin secretion [105,106,107], and liver cirrhosis [108,109,110,111,112], consequently mitigating associated risk factors. Therefore, the development of specific KP enzyme inhibitors is highly desirable (Figure 3).

##### Hepatic Fibrosis

IDO1 has been identified as a significant factor in the development of hepatic fibrosis and a potential therapeutic target [113]. Danshensu (DSS), a water-soluble active component of Salvia miltiorrhiza extract commonly used in traditional Chinese medicine, exhibits various pharmacological activities. Studies have demonstrated its potent ability to reduce the progression of hepatic fibrosis [114]. Preclinical research has shown that intragastric administration of DSS effectively inhibits JAK2-STAT3 signaling, leading to a reduction in IDO1 expression, STAT3 phosphorylation, and nuclear localization. These findings suggest an association between the activation of IDO1 and JAK2/STAT3 pathways and the progression of hepatic fibrosis, indicating the potential utility of DSS as a complementary agent for the treatment and prevention of hepatic fibrosis [114]. Conversely, ginsenoside-Rg1, isolated from ginseng, protects against IDO1 overexpression-dependent mechanisms implicated in the progression of hepatic fibrosis in animal models. Its administration significantly reduces hepatic IDO1 expression levels, resulting in decreased hepatocyte apoptosis rates in hepatic fibrosis mouse models. Additionally, it enhances the maturation of hepatic dendritic cells, which are inhibited during this disease by IDO1-dependent activation of the kynurenine pathway in the liver [114,115,116,117]. Therefore, ginsenoside-Rg1 presents another potential therapeutic agent for the treatment of hepatic fibrosis (Figure 3).

##### Atherosclerosis

The role of IDO1 in the process of atherosclerosis is complex and not fully understood [118]. Several authors have observed discrepancies regarding its function in this process. Despite the well-established protective effect of IDO1 against established atherogenesis [118,119,120], it also exhibits pro-atherosclerotic functions during the developmental stages of atherosclerosis [93,118,121,122]. Yun et al. demonstrated that IDO1 activation leads to the modulation of T-cell responses, providing atheroprotection and enhancing plaque stability [123]. Liang et al. found that the expression and activity of IDO1, as well as TDO, increase with the histological grade in early atherosclerosis [118]. Conversely, the inhibition of IDO1 using 1-MT inhibited the development of atherogenesis in high-fat diet-fed, atherosclerosis-prone apolipoprotein E-deficient mice. Therefore, the administration of IDO1 inhibitors may prove to be an effective preventive and therapeutic strategy for the early stages of atherosclerosis (Figure 3).

##### Acute Kidney Injury (AKI)

AKI following ischemia-reperfusion injury (IRI) is associated with high mortality and a lack of specific therapy [124,125,126]. Excessive activation of IDO1 has a pro-apoptotic effect in renal tubular epithelial cells [18,127,128], exacerbating kidney injury and impairing function [129]. Animal studies have demonstrated that inhibition of IDO1 by 1-MT alters the transcription of both coding and non-coding sequences within the IRI transcriptome [130]. These molecular changes appear to contribute to protection from the harmful consequences of renal IRI. During the recovery from renal IRI, pretreatment with 1-MT alleviates alterations in coding sequences associated with this disorder and triggers changes in non-coding transcripts, primarily represented by small nucleolar RNA [131,132]. This suggests a biological role of non-coding, IDO-dependent RNA sequences in regulating the early response to IRI, while inhibition of IDO1 may represent a novel strategy to reduce AKI following IRI progression (Figure 3).

##### Skin Wound Healing

Skin wound healing involves a complex process comprising several stages, including inflammation, proliferation, and remodeling [133]. During the inflammation phase, pro-inflammatory cytokines and chemokines are induced at the wound site, contributing to wound healing development [134,135]. These factors also induce the synthesis and activation of IDO1 [136,137]. However, overactivity of this enzyme may prolong the wound healing time, while its inhibition accelerates the skin wound healing process [138]. Therefore, local administration of 1-MT or other IDO1 inhibitors on the skin, such as in the form of an ointment, may be an effective approach in treating difficult-to-heal wounds, such as diabetic and pressure ulcers (Figure 3) [139].

Kynurenine (KYN) and kynurenic acid (KYNA) have also been demonstrated to be effective in preventing scar formation. Local antifibrogenic therapy using KYN is particularly appealing. However, its application is significantly limited due to its ability to cross the blood–brain barrier (BBB), potentially causing complications such as the death of excitatory neurons. Therefore, increasing attention is being directed towards its metabolite, KYNA, which does not penetrate the BBB and thus does not induce adverse effects on the central nervous system. Studies have shown that KYNA significantly increases the expression of matrix metalloproteinases (MMP1 and MMP3), while inhibiting the production of type I collagen and fibronectin by fibroblasts in vitro, without negatively impacting skin cell viability. Topical application of KYNA-containing cream to fibrotic rabbit ears specifically reduced the scar elevation index and tissue cellularity in KYNA-treated wounds compared to controls. Additionally, wounds treated with KYNA exhibited reduced levels of collagen deposition, a significant decrease in the expression of type I collagen and fibronectin, and increased expression of MMP1 compared to untreated wounds or those treated with non-active cream. These findings provide compelling evidence that KYNA represents a promising antifibrotic candidate to improve healing outcomes in patients predisposed to hypertrophic scar formation [140]. KYNA is a promising compound for the topical treatment of keloid and hypertrophic scars, as evidenced by studies in healthy adult men and women with mature keloid scars. Application of 0.5% KYNA cream twice daily led to a significant reduction in mean Patient and Observer Scar Assessment Scale (POSAS) scores after 30 days of treatment, and this improvement was maintained post-treatment. Therefore, topically applied KYNA is a potentially novel and effective method for treating mature keloid scars [141].

The involvement of the kynurenine pathway (KP) in the wound healing process is corroborated by metabolomics studies conducted on the serum of patients with chronic venous leg ulcers (CVLUs). These patients demonstrated an increase in KYN levels compared to those in the recovery phase. There was moderate support (Bayes factor = 3.70) for a negative association between changes in KYNA concentration and a linear healing slope. The results suggest that KYN and tryptophan (TRP) may be healing markers in individuals with CVLU [142]. Additionally, KYNA inhibited IL-17 and IL-23 production in vitro in CD4+ T cells and dendritic cells (DCs) through activation of G protein-coupled receptor 35 (GPCR35). This indicates that KYNA may potentially be used as an immunomodulatory agent in the treatment of IL-23- and IL-17-dependent autoimmune diseases, such as psoriasis [143].

##### Intestinal Disorders

The kynurenine pathway (KP) also plays a crucial role in maintaining intestinal well-being, with changes in the synthesis of kynurenine (KYN) and its metabolites associated with intestinal disorder development [94]. Baseline expression of IDO1 in antigen-presenting cells within the intestinal wall contributes to immune tolerance [144,145], and this enzyme can impact metabolic health by shaping intestinal microbiota [146,147]. Upregulation of IDO1 is observed during intestinal inflammatory disorders, including human inflammatory bowel diseases [112,148]. Due to the anti-inflammatory and immunosuppressive properties of KYN, pharmacological agents capable of, depending on the need, inhibiting or potentiating expression or activation of IDO1 have the potential to be used in treating various intestinal diseases, such as colitis or inflammatory bowel disease (Figure 3) [149,150]. It has been proven that activation of the IDO pathway may be harmful to intestinal inflammation in mice and humans [151]. A meta-analysis of transcriptomic datasets showed that genes involved in TRP metabolism are upregulated in Crohn’s disease (CD) and ulcerative colitis (UC) and return to baseline after successful infliximab treatment. Microarray and mRNAseq profiles from multiple experiments confirmed that the enzymes responsible for TPR degradation in the kynurenine pathway (IDO1, KYNU, KMO, and TDO2), the TRP metabolite receptor (HCAR3), and the enzymes catalyzing NAD+ turnover (NAMPT, NNMT, PARP9, CD38) were synchronously coregulated in IBD but not in intestinal cancer [152]. Chemotherapeutic drug-induced intestinal damage was often characterized by a rapid increase in TRP-KYN axis metabolism. KYNA resulted in the formation of a positive feedback loop with the IL-6-IDO1-AhR hydrocarbon receptor. Vardenafil and linagliptin as GPR35 and AHR agonists, respectively, significantly attenuated chemotherapy-induced intestinal toxicity in vivo, suggesting that chemotherapeutics in combination with them may represent a promising therapeutic strategy for cancer patients in the clinical setting [153].

However, literature data regarding IDO1 as a potential therapeutic target are ambiguous. Intestinal microbe-derived Lys has been shown to promote IDO1 expression in colonic DCs. By administering Dub or Lys, it is possible to restore the rebalance of Treg/Th17 responses and thus protect mice from colitis [154].

#### 4.1.2. Activation of IDO

##### Idiopathic Pneumonia Syndrome

The lungs have evolved mechanisms to mitigate the severity of immunopathological processes during immune responses [147]. This is crucial for maintaining lung function during inflammatory conditions and ensuring survival. Immunosuppressive mechanisms in the lungs appear to be linked to kynurenine pathway (KP) activation (Figure 3) [18,155]. Lee et al. demonstrated that donor CD4+ T cells transiently induce IDO expression in lung parenchyma in an IFN-γ-dependent manner after allogeneic hematopoietic stem cell transplantation (HSCT). Inhibition of host IDO expression, achieved by suppressing IFN-γ synthesis, leads to acute lethal pulmonary inflammation, known as idiopathic pneumonia syndrome (iPS) [156]. Interestingly, IL-6 can induce IDO expression potently in an IFN-γ-independent manner when STAT3 deacetylation is inhibited [156,157]. Treatment with a histone deacetylase inhibitor (HDACi) prevents the downregulation of IDO caused by IFN-γ inhibition, but only in an IL-6-dependent manner. Kynurenine produced by lung epithelial cells and alveolar macrophages during iPS progression suppresses the pro-inflammatory activities of lung epithelial cells and CD4+ T cells through the AhR pathway [156,158]. Consequently, HDACi can inhibit iPS development when IFN-γ expression is suppressed [156]. This underscores IDO’s critical role as a regulator of acute pulmonary inflammation and suggests that upregulation by HDACi could be a therapeutic approach for iPS after HSCT. Therefore, the addition of HDACi, such as SB939, to the immunosuppressive protocol might be an effective strategy for preventing and treating iPS after HSCT [156].

### 4.2. Pharmacological Inhibition of KMO Activity

#### 4.2.1. Acute Kidney Injury

Apart from IDO1, KMO is also highly expressed during AKI [159,160]. In an experimental AKI model induced by kidney ischemia-reperfusion injury (IRI), inhibiting KMO activity using GSK065, GSK366, and GSK428, as well as downregulating its expression, preserved renal function and reduced renal tubular cell injury. These findings indicate that increased kynurenine pathway (KP) flux through KMO contributes to the severity of AKI after IRI [161,162]. Additionally, inhibiting KMO activity during AKI appears to be a more specific and safe solution than targeting IDO, making it a valuable therapeutic approach to protect against AKI development caused by acute inflammation (Figure 3) [160,163].

#### 4.2.2. Acute Pancreatitis

Acute pancreatitis (AP) is a devastating sterile inflammation that can lead to systemic multiple organ dysfunction syndrome (MODS), often resulting in death [164,165,166]. Acute mortality from AP-MODS exceeds 20%, and even survivors often have a shortened lifespan. Currently, there is no specific therapy to protect against symptoms of acute pancreatitis with MODS (AP-MODS) [167]. Recent observations suggest that KMO activity and 3-HKYN plasma concentration positively correlate with inflammation levels, the incidence of organ dysfunction, and the severity of AP and AP-MODS courses [167,168]. This indicates that KMO overactivity may be a metabolic mechanism contributing to organ injury triggered by sterile initiators of systemic inflammation. GSK180, a potent and specific KMO inhibitor, has demonstrated therapeutic protection against AP-MODS in experimental models of AP [167,169]. These findings underscore the therapeutic potential of KMO inhibitors in treating critical diseases and suggest them as a potential therapeutic approach in managing AP and intervening early against AP-MODS (Figure 3).

#### 4.2.3. Sickness Behavior

Sickness behavior, triggered by immune system activation, is among the organism’s strategies to combat infections [170,171]. It is induced by bacterial endotoxins, such as LPS, which stimulate the release of pro-inflammatory cytokines, leading to KP activation and behavioral alterations. KP metabolites play a significant role in mediating sickness-like behavior and neuroinflammation induced by LPS [172,173,174]. Omega-3 polyunsaturated fatty acids (PUFAs) found in fish oil (FO) have anti-inflammatory properties [175,176,177,178,179]. FO has been shown to effectively inhibit KP dysregulation and mitigate sickness behavior induced by LPS in aged mice. Moreover, FO administration notably blocks LPS-induced activation of IDO and KMO within brain tissue, resulting in decreased brain levels of KYN and 3-HKYN [172]. These beneficial effects of PUFAs may be associated with their ability to modulate central inflammation, KP activity, and serotonergic signaling, as well as to incorporate into neuronal membranes [180,181]. Therefore, the PUFAs present in FO appear to be ideal candidates for nutritional interventions to decrease KP activation and neuroinflammation levels, which are associated with several neuropsychiatric and neurodegenerative diseases affecting elderly individuals (Figure 3) [182].

#### 4.2.4. Intestinal Disorders

Mice with dextran sulfate sodium (DSS)-induced colitis showed impaired TRP metabolism along with upregulation of KMO and kynureninase (KYNU). These results were confirmed by studies conducted in patients with active UC, in which both KMO and KYNU expression were positively correlated with the inflammatory factors TNF and IL-1β. It was observed that pharmacological blockade of KMO or genetic silencing of KYNU suppressed the expression of pro-inflammatory cytokines triggered by IL-1β in intestinal epithelial cells. Moreover, blocking KMO with the selective inhibitor Ro 61-8048 alleviated the symptoms of DSS-induced colitis in mice, which was accompanied by an increase in the NAD+ pool and restoration of the redox balance. This study provides evidence of the pro-inflammatory properties of KMO and KYNU in the treatment of intestinal inflammation, indicating a promising therapeutic approach in the treatment of ulcerative colitis by targeting these enzymes (Figure 3) [183].

## 5. Infectious Diseases

Numerous studies have demonstrated that the activation of the kynurenine pathway (KP) mainly plays a harmful role in the development and progression of several parasitic [184,185,186,187,188], bacterial [189,190,191,192,193], and chronic viral infections [194,195,196,197,198] (Figure 3).

### 5.1. Pharmacological Inhibition of IDO1 Activity

#### 5.1.1. Malaria

IDO1 is involved in the pathogenesis of cerebral malaria, the most severe and often fatal neurological complication of Plasmodium falciparum infection [199,200,201]. N-aryl-9-aminobenzo[b]quinolizinium derivatives, effective inhibitors of IDO1, have demonstrated activity against P. falciparum in cell culture. These compounds directly affect parasite growth and simultaneously inhibit IDO1 activation in the host [202]. Furthermore, their administration significantly reduces the severity of systemic symptoms of cerebral malaria in animal models [202]. Therefore, they may effectively complement the treatment of patients suffering from this disease.

#### 5.1.2. In Utero Bacterial Infection

Tryptophan (TRP) catabolism via the KP is upregulated in the human placenta in response to in utero infection, leading to increased release of pro-inflammatory and neuroactive factors into the fetal circulation [203,204,205,206,207]. Lipopolysaccharide (LPS), a bacterial factor, triggers KP activation and increased secretion of pro-inflammatory cytokines within the placenta [203,207,208]. Sulfasalazine, an anti-inflammatory and immunosuppressive drug commonly used in rheumatoid arthritis and ulcerative colitis treatment [209,210], effectively inhibits LPS-dependent nuclear factor-kB activation and release of pro-inflammatory cytokines in the placenta, leading to decreased IDO expression [211]. These observations suggest that modulation of KP activity in the human placenta, particularly IDO, may be a new potential therapeutic target for managing in utero infections.

#### 5.1.3. Influenza Infection

Viral infections induce tryptophan depletion and kynurenine accumulation due to increased IDO activity in infected cells to suppress the immune response against them [212]. In animal models, IDO expression is upregulated in lung tissue immediately after influenza infection [213,214], leading to immunosuppression within this tissue by decreasing the synthesis of pro-inflammatory cytokines and T-cell activity. Treatment with 1-methyl-tryptophan (1-MT) increases the expression of pro-inflammatory factors and enhances immune cell response [215], potentially limiting initial damage caused by the influenza virus in nasal and bronchial epithelial cells. Additionally, inhibiting IDO1 leads to increased T cell adaptive immune response to influenza infection [216], suggesting that controlling IDO1 activity during influenza vaccination may increase efficacy and robustness of the T cell response, improving influenza-specific heterosubtypic immunity. Interestingly, antiviral compounds inhibiting influenza virus entry prevent KP activation [197], whereas agents blocking later infection stages stimulate this pathway. Therefore, since kynurenine has been implicated as a factor triggering pain hypersensitivity during viral infection, antivirals inhibiting virus entry or a combination of antivirals impacting later stages of influenza virus infection with IDO1 inhibitors may become a novel strategy for treating and mitigating severe influenza cases with accompanying neurological symptoms [217].

#### 5.1.4. HIV Infection

Similar to influenza infection, in vivo research has shown that human immunodeficiency virus (HIV)-1 infection induces IDO overactivity [218]. This phenomenon is mostly observed in human monocyte-derived macrophages. The role of macrophages in HIV-1 infection appears to involve a complex interplay between numerous inflammatory factors and the virus. The primary mechanism involved in IDO activation by HIV-1 seems to be associated with a direct effect of the virus or its proteins rather than an indirect induction by IFN-γ or other pro-inflammatory cytokines [219]. However, it is also possible that some cellular factors may partly contribute to KP upregulation during the development of HIV-1 infection. Increased KP activity exacerbates mechanisms that suppress the immune response [12]. Among the enzymes in this pathway, IDO1 has been recognized as a significant immune response checkpoint, playing an important role in HIV-1 infection-associated immune dysfunction, even in the context of antiretroviral therapy [220]. This suggests that an increased level of immunosuppressive KP metabolites is one of the mechanisms that facilitates the development and progression of this infection and may limit the effectiveness of its treatment. This is supported by the consistent association of KP activity with decreased CD4+ T cell counts, elevated T cell activation, and viral load. Additionally, it independently predicts mortality and morbidity from events unrelated to acquired immunodeficiency syndrome (AIDS) [221]. A better understanding of KP dysregulation induced by HIV-1 may provide more selective and effective therapeutic solutions against HIV infection and AIDS development. Furthermore, broader knowledge concerning KP modulation may lead to new pharmacological approaches for AIDS-associated dementia and other neurological inflammatory disorders associated with macrophages or microglia.

#### 5.1.5. Tuberculosis

In addition to HIV-1, IDO-mediated tryptophan (TRP) catabolism via the KP is commonly enhanced in the course of tuberculosis [222]. A cumulative overactivity of IDO induced by both HIV-1 and Mycobacterium tuberculosis appears to be responsible for the extraordinarily high rate of progression to active tuberculosis observed in HIV-1-infected patients [223]. However, it remains unclear whether the increase in IDO activity is causative of progression to active disease or a compensatory response to the tuberculosis microbe. Nevertheless, assessing the systemic level of KP metabolites seems to be a promising diagnostic method for detecting active tuberculosis and monitoring the effectiveness of its treatment. Moreover, in vivo research has indicated that inhibiting IDO activity has the potential to be an effective and clinically relevant host-directed approach for tuberculosis treatment [224]. Therefore, confirmatory studies are necessary to better understand the role of KP in the pathogenesis of tuberculosis and confirm the utility of IDO inhibition as a potential therapeutic target for tuberculosis.

### 5.2. Pharmacological Inhibition of KMO Activity

#### 5.2.1. Trypanosomiasis and Malaria

Preclinical studies have shown that treatment with Ro 61-8048, a potent, high-affinity pharmacological inhibitor of KMO, significantly decreases the severity of neuroinflammatory responses in mice exhibiting severe, late central nervous system (CNS) stages of African trypanosomiasis [225]. This effect was not observed during the early stages of the disease. In vitro analysis has indicated that this compound does not directly inhibit trypanosome proliferation, suggesting that the observed anti-inflammatory effect results from its impact on host cells. This effect appears to be associated with the increase in the level of anti-inflammatory KP metabolites observed after KMO inhibition, such as KYNA and anthranilic acid (AA), which induce the expression of chemotactic monocyte chemoattractant protein 1 [226]. These findings also suggest that 3-hydroxykynurenine (3-HKYN) and its metabolites are involved in the development of severe inflammatory responses associated with the late CNS stages of trypanosomiasis. They further suggest that KMO inhibitors co-administered with conventional pharmacotherapy may be effective in preventing or ameliorating post-treatment reactive encephalopathy [227]. Moreover, Ro 61-8048 prevents death and ataxia in mice infected with Plasmodium falciparum [228], indicating the potential utility of KMO inhibitors in the treatment of both trypanosomiasis and malaria.

#### 5.2.2. SIV/HIV Infection

Swainson et al. demonstrated that KMO inhibition using CHDI-340246 in the acute simian immunodeficiency virus (SIV) infection rhesus model prevents an increase in KP downstream metabolite levels accompanying the infection [221]. As a result, it improves further clinical outcomes of this disease, reflected in an increase in CD4+ T cell count and body weight. Additionally, inhibition of KMO significantly increases the number of naive T cells and downregulates programmed cell death protein 1 (PD-1) expression in naive and memory T cell subpopulations. Importantly, early PD-1 overexpression during acute SIV infection is associated with worse clinical outcomes [221,229]. Therefore, KMO inhibition in early acute SIV/HIV-1 infection may provide significant clinical benefits [221]. It could be effective as an adjunctive treatment in SIV/HIV-1 infection to slow disease progression and improve immune reconstitution.

## 6. Other Disorders

### 6.1. Pharmacological Inhibition of IDO Activity

#### 6.1.1. Endometriosis

Elevated expression of IDO1, observed in endometrial stromal cells (ESCs), may indirectly promote the synthesis of cyclooxygenase-2 (COX-2) and matrix metallopeptidase 9 (MMP-9). These pro-inflammatory factors induce abnormal growth of ESCs and initiate invasion and implantation of shed endometrium to the peritoneum, leading to the formation of endometrial lesions and anomalous endometrium decidualization [230]. Additionally, IDO1 may activate immunosuppressive macrophages, which can facilitate the survival of endometrial tissues and seem to be involved in the progression of endometriosis. Inhibition of IDO1 by 1-MT or its L isoform suppressed the expression of COX-2 and MMP-9 in ESCs, contributing to a decrease in their proliferation, adhesion, and invasion [230,231]. These data indicate that 1-MT or other IDO1 inhibitors can be applied to prevent the development of endometriosis (Figure 3) [232]. However, further studies are required to identify IDO1-targeting therapies that do not affect hormonal balance and are safe and clinically relevant.

#### 6.1.2. Other Disorders

Galanal, a labdane-type diterpene derived from Myoga flowers, competitively inhibits IDO1 activity [233]. Even very low doses of this compound potently block the enzymatic activity of IDO1, as well as downregulate its mRNA expression. The mechanism of galangal is associated with the disruption of the transcriptional functions of nuclear factor-kB and the IFN-γ signaling pathway, which is responsible for the induction of IDO1 expression [234]. These properties of galangal make it a potentially valuable immunomodulatory agent [235]. Galanal has stronger inhibitory potency than 1-MT; therefore, this compound has great potential as a novel drug for the treatment of various inflammatory disorders associated with potent IDO1 overactivity, including immune-related diseases or tumors (Figure 3) [234].

## 7. The Beneficial and Harmful Effects of KP Activation

For the beneficial effects of KP activation in autoimmune diseases, TRP metabolites such as KYN and KYNA are responsible for immunosuppression. Their primary role is to suppress overactive immune responses. By promoting regulatory T cells (Tregs) and inhibiting effector T cells, these metabolites help alleviate the excessive immune activity characteristic of autoimmune diseases, thereby reducing inflammation and tissue damage. The activation of IDO in response to inflammation, leading to the conversion of TRP into KYN, plays an important role in these beneficial effects. As a result of reducing TRP levels, T lymphocyte proliferation is inhibited, which may help control autoimmune reactions and maintain immunological tolerance [12]. Moreover, in autoimmune diseases involving the central nervous system, such as multiple sclerosis, the neuroprotective properties of KYNA may be particularly beneficial. By antagonizing excitotoxic amino acid receptors, KYNA helps protect neurons against damage caused by inflammatory processes.

It should also be considered that KP activation in autoimmune diseases leads to increased concentration of quinolinic acid (QA), a neurotoxic compound [47,236]. Elevated QA levels may contribute to neurodegeneration and cognitive disorders in autoimmune diseases, posing a serious problem in multiple sclerosis, where both inflammation and KP dysregulation are involved in neuronal damage. Another consequence is excessive immune suppression, which increases the risk of infection and may hinder the body’s ability to eliminate cancer cells. Additionally, KP dysregulation has been associated with psychiatric symptoms, including depression and anxiety, which are common comorbidities in autoimmune diseases. Elevated levels of certain KP metabolites may disrupt the balance of neurotransmitters, contributing to mental health problems.

## 8. Novel Therapeutic Strategies Aimed at Selectively Modulating KP Activity

Designing therapeutic strategies to selectively modulate kynurenine pathway (KP) activity without potentiating deleterious effects involves a multifaceted approach that targets specific enzymes, receptors, and metabolic pathways within the KP. This precise action aims to provide beneficial effects such as immune regulation and neuroprotection while minimizing adverse effects like neurotoxicity and excessive immune suppression.

Selective inhibition of specific enzymes in the KP can help modulate the production of beneficial and harmful metabolites. The primary focus is on inhibiting IDO activity to limit immunosuppressive effects and potentially enhance anti-tumor immunity [237]. However, it is important to note that excessive inhibition may lead to undesirable inflammatory reactions. The therapeutic potential of selectively modulating IDO1 activity lies in a multifaceted approach that combines enzyme inhibition [238], modulation of downstream pathways [239], advanced drug delivery systems [240], and personalized medicine strategies [241]. These approaches aim to maximize the beneficial effects of IDO1 modulation, such as immune regulation and neuroprotection, while minimizing adverse outcomes like neurotoxicity and excessive immune suppression.

Another critical enzyme is KMO, which converts KYN into 3-HKYN. Increased KMO activity leads to the production of QA, a neurotoxic compound. Therefore, inhibiting KMO can lower QA levels, mitigating neurotoxicity while preserving the production of the neuroprotective metabolite KYNA [49].

Increasing the levels of neuroprotective and immunomodulatory KP metabolites is another therapeutic approach. This can be achieved through the use of synthetic analogs or by promoting the activity of kynurenine aminotransferases (KATs), which may provide neuroprotection and reduce excitotoxicity in diseases such as multiple sclerosis [15].

Using advanced drug delivery systems to target specific tissues or cells may improve the effectiveness and safety of KP-modulating therapies. Encapsulating KP modulators in nanoparticles or liposomes can enhance their delivery to specific sites, such as inflamed tissues or the central nervous system, thereby reducing systemic side effects [240].

Combining KP modulators with other therapeutic agents can provide a synergistic effect, improving therapeutic outcomes while minimizing side effects. For instance, co-administering KP inhibitors with anti-inflammatory drugs may help control autoimmune reactions without excessively suppressing the immune system. In cancer therapy, combining IDO inhibitors with immune checkpoint inhibitors (e.g., PD-1/PD-L1 inhibitors) may increase anti-tumor immunity by reducing immune tolerance mechanisms [10].

Tailoring therapy based on individual metabolic and genetic profiles can optimize therapeutic outcomes and minimize side effects. Developing and using biomarkers to monitor KP activity and response to therapy may help personalize treatment plans. This would enable patients to receive the most appropriate and effective therapy based on their specific KP dynamics [242].

Modulating receptors involved in the signaling of KP metabolites is another promising approach. The use of NMDA receptor antagonists may help alleviate QA-induced neurotoxicity [243]. This approach could be particularly beneficial for diseases in which QA levels are elevated.

Thus, designing therapeutic strategies to selectively modulate KP activity is promising. However, due to the complexity of KP, involving numerous enzymes and metabolites with opposing activities, it requires precise targeting and careful management. Addressing these challenges is crucial to developing effective and safe treatments that will leverage the therapeutic potential of KP while minimizing harmful effects.

Although several key KP metabolites, such as KYN and KYNA, are known to play a role in immune regulation, the full spectrum of KP metabolites and their specific immune regulatory functions remain poorly understood. The specific cellular targets and signaling mechanisms by which KP metabolites exert their immunomodulatory effects are not fully elucidated. For example, the interaction of KP metabolites with receptors such as the AhR and subsequent effects require further investigation. The role of KP in various disease contexts, including autoimmune diseases, cancer, and neurodegenerative diseases, requires further research. A key gap is the differential activation of KP in these diseases and its impact on disease progression and response to treatment. There is limited understanding of the longitudinal effects of KP activation on the immune system and its systemic effects, including potential neurotoxic or neuroprotective effects. Long-term studies are needed to elucidate these effects. Filling these research gaps with targeted experimental approaches will significantly advance the understanding of KP in immune regulation. Application of multi-omics to comprehensively map KP metabolites and assess their impact on immune regulation, cell-specific knockout models to identify specific cellular targets and mechanisms of action of KP metabolites, disease-specific animal models, and longitudinal studies to understand the long-term and systemic effects of activation KP can provide comprehensive insight into the roles and mechanisms of KP, paving the way for new therapeutic strategies.

## 9. Future Directions

Among the compounds capable of modulating the activity of the KP, inhibitors of IDO1 have become an important class of pharmaceuticals. This is because kynurenine (KYN) and its metabolites participate in numerous physiological and pathological processes [1,95,244], such as tumor immune escape [245], development of infectious [207,220,222], autoimmune [50,68,105], or metabolic disease [91,92,94], and neurodegeneration [5,22]. They are mostly considered therapeutics for cancer therapy, mainly in combination with conventional treatments, immunogenic chemotherapy, or immune checkpoint drugs, where they have shown decent effectiveness and high safety levels [10,24,25]. In addition to antitumor therapies, IDO1 inhibitors seem to be an effective solution in the management of various inflammatory disorders [131,215,216], including autoimmunological diseases [60,61,67].

In support of the above, preclinical studies have demonstrated that inhibition of IDO1 can be effective in combination with B cell depletion therapy in the treatment of RA [61], as well as in managing SLE-associated fatigue [246], morphea [77], and other cutaneous sclerosing disorders [247]. IDO1 overactivity has also been observed in the course of viral diseases [214]. In animal models, its expression was upregulated in lung tissue immediately after contracting the influenza virus [217], as well as in monocyte-derived macrophages during HIV-1 infection [219]. Therefore, IDO1 inhibitors may become a novel strategy for treating and mitigating their symptoms. They have great potential as immune adjuvants in combination with influenza vaccine and antiretroviral therapy [214,220]. Apart from infectious diseases, modulation of IDO1 activity has potential in treating various intestinal diseases, such as colitis or inflammatory bowel disease [112]. Furthermore, the activation of IDO1 also negatively affects the rate of wound healing, so the use of IDO1 inhibitors may be an effective approach to treating difficult-to-heal wounds [139]. Other research indicates that they could be administered to prevent endometriosis [231], as well as to slow the progression of atherosclerosis [121] and hepatic fibrosis [113]. Additionally, the downregulation of KP activity, especially IDO, in the human placenta may help manage in utero infections [207].

It is worth noting that the authors indicate the presence of some controversy associated with the use of IDO inhibitors. Competitive inhibitors, such as 1-MT, commonly used in preclinical research, have demonstrated low effectiveness in in vivo conditions because they only reach therapeutic effects at high serum concentrations, comparable to TRP [10]. Furthermore, they meet numerous structural and functional criteria as potential agonists of the AhR with similar KYN binding affinity. Therefore, their low systemic immunomodulatory effect may be associated with their off-target effects, such as AhR activation [248]. This aspect should not be overlooked when assessing their clinical outcomes and potential utility. Moreover, the inhibition of IDO1 activity does not always prove to be therapeutically desirable [249]. Activation of this enzyme seems to be a promising approach to managing certain inflammatory disorders. This solution turned out to be effective in an animal model of autoimmune diabetes [250] and appears to be useful in preventing and treating idiopathic pneumonia syndrome after HSCT [156]. This all underscores the necessity for a better understanding of the basic biological role of IDO1, IDO2, and TDO to develop new, more effective, and safer therapeutic strategies, as well as to identify patients who will benefit the most from their use. To date, no clinical trials have been conducted on compounds modulating the activity of KP enzymes in non-neoplastic diseases.

Beyond the pharmacological modulation of IDO1 activity, the authors propose other approaches that may serve as therapeutic targets for inflammatory disorders with abnormal TRP catabolism. TDO and IDO2 inhibition may decrease KYN production when inhibition of IDO1 is not sufficient to achieve the desired therapeutic effect [251]. Targeting more than one enzyme at once may induce a wide and synergistic systemic response, which could be especially effective in patients with inherent or acquired resistance to currently available inhibitors of IDO1. Simultaneous targeting of IDO2 and IDO1, though less understood, may provide new therapeutic options for managing inflammatory disorders, including autoimmune ones. Moreover, the idea of dual IDO1 and TDO inhibitors is also desirable, as inhibition of IDO may lead to the activation of TDO [10]. However, all these enzymes belong to upstream levels of the KP, and their inhibition may cause more adverse effects than selective approaches.

Targeting KMO and KAT activity appears to be an effective solution when inhibiting upstream enzymes in the KP is not desirable. Preclinical studies have found that pharmacological inhibition of KMO activity may find application in treating certain diseases [161,163,165]. In some cases, blocking this enzyme seems to be a better therapeutic approach than targeting IDO. Since KMO overactivity has been shown to contribute to the severity of AKI after IRI, AP, and AP-MODS symptoms, administering KMO inhibitors appears to be an effective strategy for managing these disorders [162,166,167]. Additionally, these compounds seem to be potentially useful in treating some parasitic diseases, including trypanosomiasis and malaria [199,200]. The authors also showed that KMO inhibition can be effective as an adjunctive treatment for SIV and HIV infections [221]. It slows the progression of these diseases and enhances immune reconstitution. In turn, KAT inhibitors might be effective in managing endotoxic shock [252]. Apart from that, they have the potential to be a novel therapeutic solution in treating pre-eclampsia and other cardiovascular diseases. PF-04859989 is a new, potent, highly specific, and irreversible inhibitor of the human and rat KAT II isoform [253,254]. It could be useful in treating these disorders, as well as others accompanied by excessive activation of this enzyme. However, many unknowns remain concerning this issue, as the expression profile and pathophysiological role of KAT isoforms in the cardiovascular system remain unclear, requiring further research to resolve them.

In addition to directly modulating KP enzymes, the authors suggest other therapeutic targets that could address inflammatory disorders associated with abnormal KP activity. These include modulating the expression levels of IDO, TDO, and KMO, regulating the proteasomal degradation rate of these enzymes, and targeting upstream factors responsible for the synthesis and activation of IDO, such as the JAK/STAT3 pathway, or downstream effectors for KP metabolites, including general control nonderepressible-2 kinase and mammalian target of rapamycin kinase [65,114]. Additionally, direct targeting of specific KP metabolites using monoclonal antibodies, as well as inhibiting receptors for KP metabolites such as AhR and G protein-coupled receptor 35, or mechanisms responsible for TRP cellular transport, should also be considered as potential therapeutic solutions [10,166]. These approaches may mitigate the detrimental effects of KP metabolites, regardless of their source and synthesis level. AhR antagonists, in particular, have shown promising results in animal models in this regard [255,256,257].

## 10. Conclusions

Physiologically, the KP plays an important role in maintaining the balance of the immune system. However, various inflammatory conditions are accompanied by its excessive activation, leading to the accumulation of its active metabolites. This accumulation may adversely affect numerous biological processes, intensifying disturbances in systemic homeostasis caused by the ongoing disease. Pharmacological modulation of KP activity appears to be an effective solution for the adjunctive treatment of many inflammatory disorders, including autoimmune conditions. Although inhibition of this pathway is considered a therapeutic target in most assessed disorders, this approach is not always desirable. Depending on the nature of the disorder, activation of the pathway may bring beneficial effects in some cases. Among the enzymes belonging to the KP, IDO, TDO, and KMO show the greatest potential as therapeutic targets in various inflammatory disorders. Results of preclinical studies suggest that modulation of their activity may provide new, effective strategies for managing numerous inflammatory diseases, including those currently considered incurable or with limited therapeutic options, such as autoimmune conditions. This underscores the importance of further research on this topic, particularly concerning their clinical application.

## Figures and Tables

**Figure 1 cells-13-01259-f001:**
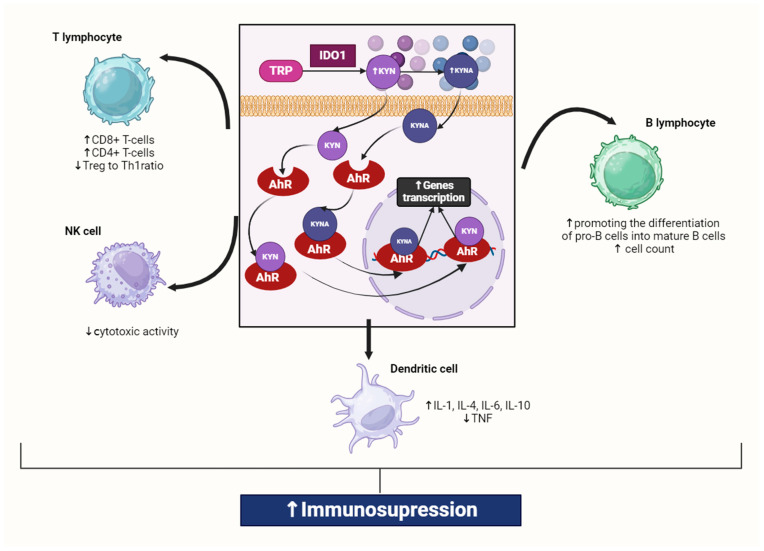
Interaction of kynurenine and kynurenic acid with AhR in macrophage. Kynurenines, produced through tryptophan degradation via IDO1, activate AhR, which is present in all innate immune cells. IDO1 exhibits high expression in various immune cell types, including macrophages, monocytes, dendritic cells (DCs), eosinophils, neutrophils, specific T cell subsets, and regulatory B cells. Activation of IDO1 expression and activity in professional antigen-presenting cells (APCs) such as DCs and monocyte-derived macrophages, as well as in other innate immune cells like natural killer cells, eosinophils, and neutrophils, results in enhancing the immunosuppressive effect and reducing local inflammation. It was created with BioRender (www.biorender.com). Abbreviations: TRP—tryptophan, IDO1—indoleamine 2,3-dioxygenase 1, KYN—kynurenine, KYNA—kynurenic acid, AhR—aryl hydrocarbon receptor, NF-κB—nuclear factor kappa-light-chain-enhancer of activated B cells, TNF—tumor necrosis factor, IL-6—interleukin 6, IL-1—interleukin 1, IL-4—interleukin 4, IL-10—interleukin 10, Treg—Regulatory T cell, Th1—T helper 1 cell, NK—natural killer cell, ↑—increase/rise, ↓—suppression/inhibition.

**Figure 2 cells-13-01259-f002:**
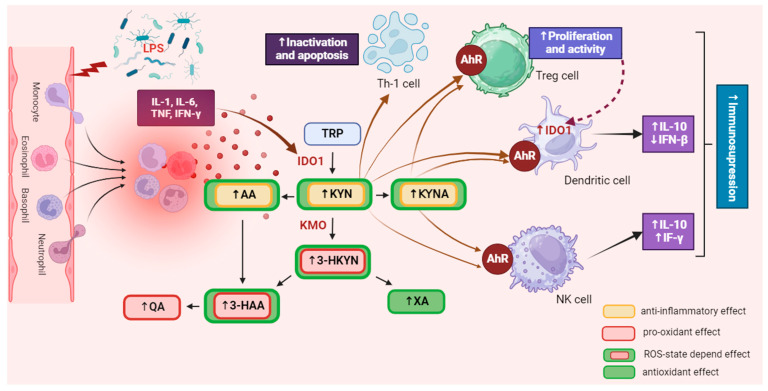
The influence of the kynurenine pathway on immune activity. IDO is extensively expressed in various immune cells, including macrophages, monocytes, dendritic cells (DCs), eosinophils, neutrophils, certain T cell subsets, and regulatory B cells. The activation of IDO expression and activity in professional antigen-presenting cells (APCs) like DCs and monocyte-derived macrophages, as well as in other innate immune cells such as NK cells, eosinophils, and neutrophils, has a diverse impact on their functions within the immune system. It was created with BioRender. Abbreviations: TRP—tryptophan, IDO1—indoleamine 2,3-dioxygenase 1, KYN—kynurenine, KMO—kynurenine 3-monooxygenase, 3-HKYN—3-hydroxykynurenine, KYNA—kynurenic acid, 3-HAA—3-hydroxyanthranilic acid, QA—quinolinic acid, AA—anthranilic acid, XA—xanthurenic acid, AhR—aryl hydrocarbon receptor, IL-1—interleukine 1, IL-6—interleukine 6, IL-10—interleukine 10, TNF—tumor necrosis factor, IFN-γ—interferon gamma, IFN-β—interferon beta, Treg—regulatory T cell, NK—natural killer cell, Th1—T helper 1 cell, ROS—reactive oxygen species. ↑—increase/rise, ↓—suppression/inhibition.

**Figure 3 cells-13-01259-f003:**
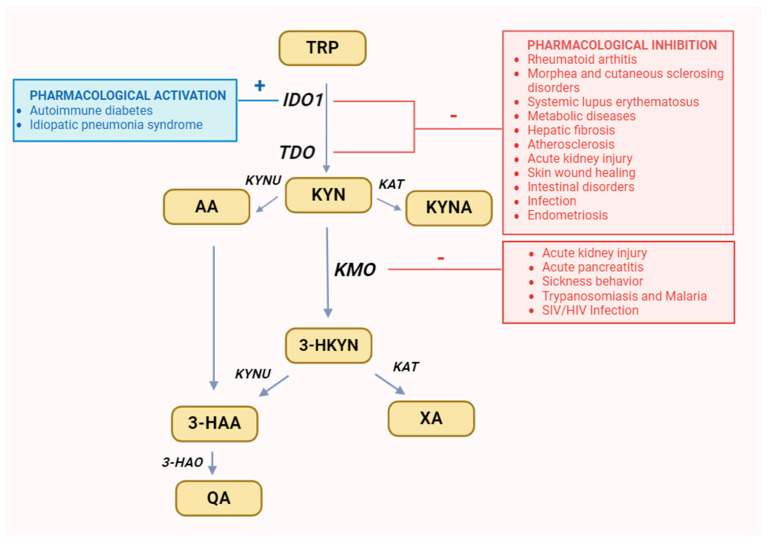
Recommended changes in enzyme activity for selected diseases. It was created with BioRender. Abbreviations: TRP—tryptophan, IDO1—indoleamine 2,3-dioxygenase, TDO—tryptophan 2,3-dioxygenase, KYN—kynurenine, KYNU—kynureninase, KAT—kynurenine aminotransferase, KMO—kynurenine 3-monooxygenase, 3-HKYN—3-hydroxykynurenine, KYNA—kynurenic acid, 3-HAA—3-hydroxyanthranilic acid, 3-HAO—3-hydroxyanthranilic acid 3,4-dioxygenase, QA—quinolinic acid, AA—anthranilic acid, XA—xanthurenic acid.

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
