# Peer review of "Kynurenines as a Novel Target for the Treatment of Inflammatory Disorders"

_cells, 2024, doi:10.3390/cells13151259_

Round 1

Reviewer 1 Report

Comments and Suggestions for Authors

The manuscript provides an extensive overview of the kynurenine pathway (KP) and its significant role in tryptophan (TRP) metabolism. The authors have highlighted the biological activities and physiological implications of KP metabolites, and the paper is well-structured, summarizing the enzymes involved in the KP and their tissue distributions. However, several points need to be addressed to enhance the clarity and impact of the manuscript:

1. Improve the balance between discussing the beneficial and harmful effects of KP activation, emphasizing the therapeutic potential and challenges of targeting the KP in disease treatment.

2. Discuss how therapeutic strategies can be designed to selectively modulate KP activity without exacerbating harmful effects.

3. Describe known regulatory mechanisms controlling KP enzyme activity under physiological and pathological conditions, including genetic, epigenetic, and environmental factors.

4. Identify the most critical research gaps in understanding KP in immune regulation, and suggest specific experimental approaches or models that could be useful.

5. Discuss how therapies can selectively enhance beneficial effects of IDO1 while minimizing adverse outcomes.

Author Response

We would like to take this opportunity to deeply thank the Reviewer who identified the parts of our manuscript that required corrections or modifications. Please find the response to the Rewiever’s comments below.

Reviewer #1:

  1. Improve the balance between discussing the beneficial and harmful effects of KP activation, emphasizing the therapeutic potential and challenges of targeting the KP in disease treatment.

Thank you for your attention. Kynurenine and kynurenic acid are metabolites in the tryptophan degradation pathway. Kynurenine is an intermediate that can be further converted into several bioactive compounds, including kynurenic acid, which is known for its neuroprotective and anti-inflammatory properties. Both compounds interact with the aryl hydrocarbon receptor (AhR), a ligand-activated transcription factor that regulates the expression of various genes involved in xenobiotic metabolism, immune response, and cellular homeostasis. The binding of kynurenine and kynurenic acid to AhR can influence numerous physiological and pathological processes, underscoring their significance in biomedical research.

Following the Reviewer's suggestion, we have added Section 7, "The Beneficial and Harmful Effects of KP Activation," to the manuscript (lines 653-676).

For the beneficial effects of KP activation in autoimmune diseases, TRP metabolites such as KYN and KYNA are responsible for immunosuppression. Their primary role is to suppress overactive immune responses. By promoting regulatory T cells (Tregs) and inhibiting effector T cells, these metabolites help alleviate the excessive immune activity characteristic of autoimmune diseases, thereby reducing inflammation and tissue damage. The activation of indoleamine 2,3-dioxygenase (IDO) in response to inflammation, leading to the conversion of TRP into KYN, plays an important role in these beneficial effects. As a result of reducing TRP levels, T lymphocyte proliferation is inhibited, which may help control autoimmune reactions and maintain immunological tolerance. Moreover, in autoimmune diseases involving the central nervous system, such as multiple sclerosis, the neuroprotective properties of KYNA may be particularly beneficial. By antagonizing excitotoxic amino acid receptors, KYNA helps protect neurons against damage caused by inflammatory processes.

It should also be considered that KP activation in autoimmune diseases leads to increased concentration of QA, a neurotoxic compound. Elevated QA levels may contribute to neurodegeneration and cognitive disorders in autoimmune diseases, posing a serious problem in multiple sclerosis, where both inflammation and KP dysregulation are involved in neuronal damage. Another consequence is excessive immune suppression, which increases the risk of infection and may hinder the body's ability to eliminate cancer cells. Additionally, KP dysregulation has been associated with psychiatric symptoms, including depression and anxiety, which are common comorbidities in autoimmune diseases. Elevated levels of certain KP metabolites may disrupt the balance of neurotransmitters, contributing to mental health problems.

  1. Discuss how therapeutic strategies can be designed to selectively modulate KP activity without exacerbating harmful effects.

Thank you for your suggestion. We discussed this problem in Section 8. Novel therapeutic strategies aimed at selectively modulating KP activity

Added text. 8. Novel therapeutic strategies aimed at selectively modulating KP activity (lines 677-744)

Designing therapeutic strategies to selectively modulate kynurenine pathway (KP) activity without potentiating deleterious effects involves a multifaceted approach that targets specific enzymes, receptors, and metabolic pathways within the KP. This precise action aims to provide beneficial effects such as immune regulation and neuro-protection while minimizing adverse effects like neurotoxicity and excessive immune suppression.

Selective inhibition of specific enzymes in the KP can help modulate the production of beneficial and harmful metabolites. The primary focus is on inhibiting IDO activity to limit immunosuppressive effects and potentially enhance anti-tumor immunity [236]. However, it is important to note that excessive inhibition may lead to undesirable inflammatory reactions. The therapeutic potential of selectively modulating IDO1 activity lies in a multifaceted approach that combines enzyme inhibition [237], modulation of downstream pathways [238], advanced drug delivery systems [239], and personalized medicine strategies [240]. These approaches aim to maximize the beneficial effects of IDO1 modulation, such as immune regulation and neuroprotection, while minimizing adverse outcomes like neurotoxicity and excessive immune suppression.

Another critical enzyme is KMO, which converts KYN into 3-HKYN. Increased KMO activity leads to the production of QA, a neurotoxic compound. Therefore, inhibiting KMO can lower QA levels, mitigating neurotoxicity while preserving the production of the neuroprotective metabolite KYNA [47].

Increasing the levels of neuroprotective and immunomodulatory KP metabolites is another therapeutic approach. This can be achieved through the use of synthetic analogs or by promoting the activity of kynurenine aminotransferases (KAT), which may provide neuroprotection and reduce excitotoxicity in diseases such as multiple sclerosis [15].

Using advanced drug delivery systems to target specific tissues or cells may improve the effectiveness and safety of KP-modulating therapies. Encapsulating KP modulators in nanoparticles or liposomes can enhance their delivery to specific sites, such as in-flamed tissues or the central nervous system, thereby reducing systemic side effects [239] .

Combining KP modulators with other therapeutic agents can provide a synergistic effect, improving therapeutic outcomes while minimizing side effects. For instance, co-administering KP inhibitors with anti-inflammatory drugs may help control auto-immune reactions without excessively suppressing the immune system. In cancer therapy, combining IDO inhibitors with immune checkpoint inhibitors (e.g., PD-1/PD-L1 inhibitors) may increase anti-tumor immunity by reducing immune tolerance mechanisms [10].

Tailoring therapy based on individual metabolic and genetic profiles can optimize therapeutic outcomes and minimize side effects. Developing and using biomarkers to monitor KP activity and response to therapy may help personalize treatment plans. This would enable patients to receive the most appropriate and effective therapy based on their specific KP dynamics [241].

Modulating receptors involved in the signaling of KP metabolites is another promising approach. The use of NMDA receptor antagonists may help alleviate QA-induced neurotoxicity [242]. This approach could be particularly beneficial for diseases in which QA levels are elevated.

Thus, designing therapeutic strategies to selectively modulate KP activity is promising. However, due to the complexity of KP, involving numerous enzymes and metabolites with opposing activities, it requires precise targeting and careful management. Addressing these challenges is crucial to developing effective and safe treatments that will leverage the therapeutic potential of KP while minimizing harmful effects.

Although several key KP metabolites, such as KYN and KYNA, are known to play a role in immune regulation, the full spectrum of KP metabolites and their specific immune regulatory functions remain poorly understood. The specific cellular targets and signaling mechanisms by which KP metabolites exert their immunomodulatory effects are not fully elucidated. For example, the interaction of KP metabolites with receptors such as the AhR and subsequent effects require further investigation. The role of KP in various disease contexts, including autoimmune diseases, cancer, and neurodegenerative diseases, requires further research. A key gap is the differential activation of KP in these diseases and its impact on disease progression and response to treatment. There is limited understanding of the longitudinal effects of KP activation on the immune system and its systemic effects, including potential neurotoxic or neuroprotective effects. Long-term studies are needed to elucidate these effects. Filling these research gaps with targeted experimental approaches will significantly advance the understanding of KP in immune regulation. Application of multi-omics to comprehensively map KP metabolites and assess their impact on immune regulation, cell-specific knockout models to identify specific cellular targets and mechanisms of action of KP metabolites, disease-specific animal models, and longitudinal studies to understand the long-term and systemic effects of activation KP can provide comprehensive insight into the roles and mechanisms of KP, paving the way for new therapeutic strategies.

  1. Describe known regulatory mechanisms controlling KP enzyme activity under physiological and pathological conditions, including genetic, epigenetic, and environmental factors.

Based on the reviewer's suggestion, we constructed the text (lines 56-77).

Regulation of the enzyme activity of the kynurenine pathway (KP) involves complex genetic, epigenetic, and environmental mechanisms, both in physiological and pathological conditions. Genetic factors appear to play a key role in controlling the activity of the KP enzyme. For example, specific polymorphisms of the IDO1 and KMO genes may affect their expression level and functional activity, and thus influence the production of both neuroprotective and neurotoxic metabolites. Genetic mutations in these enzymes have been associated with a variety of diseases, including neurodegenerative and psychiatric disorders [15,16]. Epigenetic modifications such as DNA methylation and histone acetylation also regulate the expression of KP enzymes. These modifications may be influenced by environmental factors and disease states. For example, inflammation can induce IDO1 expression by activating transcription factors such as NF-κB and STAT1, which bind to the promoter regions of the IDO1 gene. Moreover, epigenetic drugs that modify histone acetylation patterns can alter the expression of KP enzymes, thereby affecting pathway activity [4,17]. Environmental factors such as stress, infection and dietary ingredients significantly influence the activity of the KP enzyme. Inflammatory cytokines such as IFN-γ and TNF can increase IDO1 and TDO levels, increasing the conversion of TRP to KYN [18]. In pathological conditions such as cancer or chronic infection, persistent activation of these enzymes can lead to immunosuppression and contribute to disease progression. End-product inhibition is another regulatory mechanism in which accumulated downstream metabolites, such as QA, can inhibit upstream enzymes, maintaining metabolic balance [19].

  1. Identify the most critical research gaps in understanding KP in immune regulation, and suggest specific experimental approaches or models that could be useful.

Based on the reviewer's suggestion, we constructed the text (lines 726-744).

Although several key KP metabolites, such as KYN and KYNA, are known to play a role in immune regulation, the full spectrum of KP metabolites and their specific immune regulatory functions remain poorly understood. The specific cellular targets and signaling mechanisms by which KP metabolites exert their immunomodulatory effects are not fully elucidated. For example, the interaction of KP metabolites with receptors such as the AhR and subsequent effects require further investigation. The role of KP in various disease contexts, including autoimmune diseases, cancer, and neurodegenerative diseases, requires further research. A key gap is the differential activation of KP in these diseases and its impact on disease progression and response to treatment. There is limited understanding of the longitudinal effects of KP activation on the immune system and its systemic effects, including potential neurotoxic or neuroprotective effects. Long-term studies are needed to elucidate these effects. Filling these research gaps with targeted experimental approaches will significantly advance the understanding of KP in immune regulation. Application of multi-omics to comprehensively map KP metabolites and assess their impact on immune regulation, cell-specific knockout models to identify specific cellular targets and mechanisms of action of KP metabolites, disease-specific animal models, and longitudinal studies to understand the long-term and systemic effects of activation KP can provide comprehensive insight into the roles and mechanisms of KP, paving the way for new therapeutic strategies.

  1. Discuss how therapies can selectively enhance beneficial effects of IDO1 while minimizing adverse outcomes.

Based on the reviewer's suggestion, we constructed the text (lines 687-692).

The therapeutic potential of selectively modulating IDO1 activity lies in a multi-faceted approach that combines enzyme inhibition [237], modulation of downstream pathways [238], advanced drug delivery systems [239], and personalized medicine strategies [240]. These approaches aim to maximize the beneficial effects of IDO1 modulation, such as immune regulation and neuroprotection, while minimizing adverse outcomes like neurotoxicity and excessive immune suppression.

Reviewer 2 Report

Comments and Suggestions for Authors

According to the authors; statement, the review submitted for evaluation is expected to address therapeutic aspects of the effects on the kynurenine pathway in inflammation.

Unfortunately, the manuscript does not fulfill this goal. The manuscript is a sum of rather arbitrarily selected publications, but it does not bring the reader closer to the announced subject matter. The authors; narrative is unclear; the construction of the work is difficult to understand and follow. The work lacks a precise statement from the authors like "take home message".

The processes depicted schematically in the figures in the manuscript are not explained in the text, and the legend is limited only to a list of abbreviations used. The figures are not self-explanatory; on the contrary, even after careful analysis, it is difficult to understand what information they provide. It is unclear whether the processes shown in the figures occur generally throughout the body or in specific compartments only.

For example, Fig. 1 shows that KYNA passes through the cell membrane - is there uptake of KYNA? Transporters for KYNA have been described in the kidney. The increase in transcription under the influence of the KYN or KYNA complex with AhR is banal information. All in all, the reader learns that KYN and KYNA act in concert in the same comportment. Do both substances really cause the same effects?

The title of Figure 2 informs about the effect of the kynurenine pathway on inflammatory processes, but it is clear from the figure that inflammatory processes ("pro-inflammatory factor") stimulate cytokine secretion and activate the kynurenine pathway. And this is the obvious fact. The figure highlights the role of the AhR receptor, and omits the anti-inflammatory and antioxidant effects of some kynurenine pathway metabolites. The reader may be confused. Similarly, analysis of Fig. 3 may lead to the conclusion that the conditions highlighted in red cause inhibition of the kynurenine pathway or conversely, that inhibition of the kynurenine pathway leads to these diseases. What is the truth? The question arises, whether these are phenomena that can be generalized or whether the events affect only selected compartments?

The term kynurenine pathway is nonchalantly used by the authors. First, the kynurenine pathway is a collection of enzymes whereas activation of these enzymes leads to the production of tryptophan metabolites. Thus, it should be precisely defined whether the subject of the review is enzymes or metabolites or both. Second, activation of the kynurenine pathway does not lead to a common pharmacological effect. Many substances with multiple mechanisms of action are formed along the kynurenine pathway, and many of these substances have opposing effects, such as quinolinic acid vs kynurenic acid. The receptor targets of these metabolites are almost completely omitted from the manuscript with the exception of AhR. Reading the review, it seems that the kynurenine pathway refers to 3 enzymes (IDO, TDO and KMO). The reader learns even less about metabolites and their effects. Yet this is the essence of the pathophysiological effects caused by these substances and, in a broader sense, by the kynurenine pathway. The authors focus quite a lot on changes at the gene level, but this does not determine the effects of kynurenine pathway metabolites.

The review focuses on the effects of several substances/drugs, but they are mostly non-selective. In many cases, their relationship to the kynurenine pathway appears to be indirect. Why the impact of selective substances was not discussed?

The manuscript pretends to review the therapeutic possibilities of targeting the kynurenine pathway.

In fact, the reader finds only a few isolated results, and they usually involve experimental studies on animal models. Whereas, IDO/TDO inhibitors have been subjected to multiple clinical trials, but these results have not been cited and systematized. Instead, a lot of space has been devoted for example to the herbal water extract (danshensu), whose composition is not defined and whose action is far from selective.

The accuracy and reliability of the data in the manuscript is highly questionable. An example is the effect of kynurenine pathway metabolites on skin wound healing. Publication on the effect of kynurenine on wound healing (DOI: 10.1371/journal.pone.0091955 ) was omitted, particularly since clinical trials based on these results were conducted in humans (DOI: 10.36849/jdd.6197). There are more important publications on the subject (DOI: 10.1089/wound.2023.0137; DOI: 10.1007/s11010- 017-2975-3 ). Similarly, there is an extensive literature on inflammatory gastrointestinal diseases that has not been cited.

In conclusion, a review paper does not benefit the reader, and may even lead to an incorrect interpretation of the phenomena outlined in it. The final conclusion about the need for further research is trivial.

Author Response

We would like to take this opportunity to deeply thank the Reviewer who identified the parts of our manuscript that required corrections or modifications. Please find the response to the Rewiever’s comments below.

Reviewer #2:

Unfortunately, the manuscript does not fulfill this goal. The manuscript is a sum of rather arbitrarily selected publications, but it does not bring the reader closer to the announced subject matter. The authors; narrative is unclear; the construction of the work is difficult to understand and follow. The work lacks a precise statement from the authors like "take home message".

Thank you for your constructive criticism. We have revised our manuscript based on the feedback from three reviewers. Specifically, we have expanded the manuscript to include additional sections (7. The beneficial and harmful effects of KP activation and 8. Novel therapeutic strategies aimed at selectively modulating KP activity) that better emphasize the role of the kynurenine pathway in the treatment of inflammatory disorders. We believe that the manuscript has greatly benefited from the reviewers' insightful comments and suggestions.

The processes depicted schematically in the figures in the manuscript are not explained in the text, and the legend is limited only to a list of abbreviations used. The figures are not self-explanatory; on the contrary, even after careful analysis, it is difficult to understand what information they provide. It is unclear whether the processes shown in the figures occur generally throughout the body or in specific compartments only.

Regarding Figure 1. We added text with an explanation role and distribution of AhR in the body to the 2. The role of the kynurenine pathway in the immune response section.

Lines 93-98: The aryl hydrocarbon receptor (AhR) is a ligand-activated transcription factor that is widely distributed in various tissues throughout the body. AhR is highly expressed in hepatocytes, where it plays a key role in the metabolism of xenobiotics and endogenous compounds. Additionally, AhR is expressed in a variety of immune cells, including T cells, B cells, dendritic cells, and macrophages, influencing the immune response and inflammation (Figure 1).

Lines 101-108: This receptor is present in lung cells, where it helps respond to pollutants in the air. Intestinal epithelial cells and immune cells in the intestine express AhR, which is involved in maintaining intestinal homeostasis and modulating intestinal microbiota. AhR is expressed in the reproductive organs, where it may influence reproductive development and function [12,18]. Neurons and glial cells in the central nervous system express AhR, indicating a role in neurodevelopment and neuroprotection. These wide distributions highlight the importance of AhRs in regulating many physiological processes and responding to environmental stimuli.

As suggested, an explanation has been added to the caption of Figure 1. Changes to the text (lines 143-150):

Interaction of kynurenine and kynurenic acid with AhR in macrophage. Kynurenines resulting from IDO-mediated tryptophan degradation activate AhR, which is present in all innate immune cells. Consequently, this activation suppresses immune responses generated by macrophages, natural killer (NK) cells, and T lymphocytes, thereby enhancing the immunosuppressive effect and reducing local inflammation.

Kynurenines, produced through tryptophan degradation via IDO1, activate AhR, which is present in all innate immune cells. IDO1 exhibits high expression in various immune cell types, including macrophages, monocytes, dendritic cells (DCs), eosinophils, neutrophils, specific T cell subsets, and regulatory B cells. Activation of IDO1 expression and activity in professional antigen-presenting cells (APCs) such as DCs and monocyte-derived macrophages, as well as in other innate immune cells like natural killer cells, eosinophils, and neutrophils, results in enhancing the immunosuppressive effect and reducing local inflammation.

For example, Fig. 1 shows that KYNA passes through the cell membrane - is there uptake of KYNA? Transporters for KYNA have been described in the kidney. The increase in transcription under the influence of the KYN or KYNA complex with AhR is banal information. All in all, the reader learns that KYN and KYNA act in concert in the same comportment. Do both substances really cause the same effects?

Literature data indicate that most AhR ligands are hydrophobic compounds readily penetrating cell membranes. It is assumed that this also applies to KYN and KYNA. Additionally, transporters such as the System L amino Acid Transporter (SLC7A5) are known to be involved in KYN transport. However, data on the penetration/transport of KYNA across immune cell membranes are sparse. Indeed, Organic Anion Transporters 1 and 3 (OAT1/3) are well known for their role in mediating renal elimination, but their involvement appears to be limited to this organ. Given the hydrophobic nature of KYNA and its low molecular weight (189.17 g/mol), this compound is assumed to penetrate cell membranes passively. The effects exerted by AhR ligands such as KYN and KYNA are the same.

We have added a description of the interaction of KYN and KYNA with AhR to section 2. The role of the kynurenine pathway in immune response (lines 108-132).

AhR activation is closely associated with a nuclear transition that relies on a sequence of positively charged amino acids known as the nuclear localization signal (NLS). The NLS consists of two segments (bipartite NLS) and is located within the conserved basic helix–loop–helix (bHLH) domain in the N-terminal part of the protein. In its un-liganded state, AhR predominantly resides in the cytoplasm, bound to a chaperone complex comprising two molecules of HSP90, and single molecules of co-chaperone p23 and hepatitis x-associated protein-2 (XAP2). Most AhR ligands are hydrophobic, including KYN and KYNA, allowing them to cross the cell membrane through simple diffusion [26]. KYN can cross cell membranes and penetrate tissue readily [4]. Additionally, kynurenine uptake in T cells is mediated by the System L Amino Acid Trans-porter SLC7A5 [27]. When a ligand binds to cytosolic AhR, it induces a conformational change that exposes the NLS, facilitating its recognition by nuclear transporters. Specifically, members of the importin (IMP) superfamily, such as IMPβ1 or its adaptor protein IMPα, can recognize the exposed NLS. Following this recognition, AhR, along with the IMPα/β1 heterodimer, is transported into the nucleus through the nuclear pore complexes (NPCs). On the nuclear side of NPCs, IMPβ1 binds to RanGTP (Ras-related nuclear protein), which leads to the release of the NLS-cargo. This process allows AhR to enter the nucleus and exert its regulatory functions &. The combination of kynurenine and AHR leads to the formation of a complex with the nuclear translocator molecule (Arnt) in the nucleus, influencing changes in gene transcription. Activation of AHR by KYN or KYNA induces FoxP3, which promotes the differentiation of naïve CD4+ cells into a Treg cell phenotype and also inhibits RORγt expression, preventing cell maturation into Th17 cells. Thus, KYN and KYNA are at the center of the immune seesaw, which can be pro-inflammatory (via Th17 cells) or anti-inflammatory (via Tregs) [4].

The title of Figure 2 informs about the effect of the kynurenine pathway on inflammatory processes, but it is clear from the figure that inflammatory processes ("pro-inflammatory factor") stimulate cytokine secretion and activate the kynurenine pathway. And this is the obvious fact. The figure highlights the role of the AhR receptor, and omits the anti-inflammatory and antioxidant effects of some kynurenine pathway metabolites. The reader may be confused.

We fully agree with the Reviewer's comment. This figure has been corrected.

Similarly, analysis of Fig. 3 may lead to the conclusion that the conditions highlighted in red cause inhibition of the kynurenine pathway or conversely, that inhibition of the kynurenine pathway leads to these diseases. What is the truth? The question arises, whether these are phenomena that can be generalized or whether the events affect only selected compartments?

Thank you for your pertinent attention. Indeed, the message it conveyed was not ambiguous. Following the reviewer's suggestion, we have modified Figure 3. The purpose of this figure is to indicate potential targets for the treatment of selected diseases. The red color indicates diseases in which enzyme inhibition is desirable, while the blue color indicates two disease entities in which stimulation of IDO activity has a potentially beneficial pharmacological effect. Although the modulation of enzyme activity concerns specific diseases, similar pharmacological interventions may also be effective in entirely different diseases.

The term kynurenine pathway is nonchalantly used by the authors. First, the kynurenine pathway is a collection of enzymes whereas activation of these enzymes leads to the production of tryptophan metabolites. Thus, it should be precisely defined whether the subject of the review is enzymes or metabolites or both. Second, activation of the kynurenine pathway does not lead to a common pharmacological effect. Many substances with multiple mechanisms of action are formed along the kynurenine pathway, and many of these substances have opposing effects, such as quinolinic acid vs kynurenic acid. The receptor targets of these metabolites are almost completely omitted from the manuscript with the exception of AhR. Reading the review, it seems that the kynurenine pathway refers to 3 enzymes (IDO, TDO and KMO). The reader learns even less about metabolites and their effects. Yet this is the essence of the pathophysiological effects caused by these substances and, in a broader sense, by the kynurenine pathway. The authors focus quite a lot on changes at the gene level, but this does not determine the effects of kynurenine pathway metabolites.

We fully agree with the Reviewer that the kynurenine pathway comprises a set of enzymes responsible for tryptophan metabolism. This literature review aimed to identify enzymes as potential therapeutic targets. In this manuscript, we focused on the activities of IDO, TDO, and KMO because these enzymes are particularly important as targets in the treatment of the inflammatory diseases discussed in this work. We acknowledge the existence of other enzymes in this pathway, such as formamidase, kynurenine aminotransferase, kynureninase, 3-hydroxyanthranilate dioxygenase, and 2-amino-3-carboxymuconate semialdehyde decarboxylase. However, in the context of the pharmacotherapeutic approach to the diseases presented in the manuscript, only the three aforementioned enzymes are significant.

Indeed, only the AhR receptor was presented in the manuscript. However, from the perspective of immunological processes, it plays the most important role. The disease entities presented in the manuscript have an inflammatory basis, which is why we focused our attention on this receptor.

Literature data indicate that using bortezomib restores IDO1 protein levels in dendritic cells (DCs) in non-obese diabetic mice by inhibiting proteasomal degradation. This DC-mediated mechanism contributes to the suppression of the immune response. In vivo, administration of bortezomib prevents the development of autoimmune diabetes through IDO1 and DC-dependent mechanisms [87]. Another disease in which activation of IDO activity has potential therapeutic effects is idiopathic pneumonia syndrome (IPS). Treatment with histone deacetylase inhibitors (HDACi) was observed to prevent IDO downregulation caused by IFN-γ inhibition, but only in an IL-6-dependent manner. During IPS progression, kynurenine produced by lung epithelial cells and alveolar macrophages inhibits the pro-inflammatory activity of lung epithelial cells and CD4+ T cells via the AhR pathway [157,158]. In summary, activation of IDO presents a promising therapeutic strategy for these two diseases. The potential application of IDO activation is discussed in the manuscript on lines 278-294 and 433-452.

The target of pharmacological interventions is the activity of individual enzymes, and their effects result in changes in the concentrations of metabolites that can modulate receptors.

Text change (lines 77-81): Research on the biological role of the KP has led authors to conclude that targeting enzymes within this pathway may be an effective method of treating inflammatory diseases, including those with limited therapeutic options or currently considered incurable. This literature review aims to identify these enzymes as potential therapeutic targets.

The review focuses on the effects of several substances/drugs, but they are mostly non-selective. In many cases, their relationship to the kynurenine pathway appears to be indirect. Why the impact of selective substances was not discussed?

We fully agree with the Reviewer that the action of the presented substances/drugs is only indirect. We did not present selective substances in the manuscript because no literature data is confirming their potential use in the discussed diseases. The primary application of selective substances, such as selective IDO inhibitors, is cancer immunotherapy, as substantiated by ongoing clinical trials (Yang, P.; Zhang, J. Indoleamine 2,3-Dioxygenase (IDO) Activity: A Perspective Biomarker for Laboratory Determination in Tumor Immunotherapy. Biomedicines 2023, 11, 1988).

The manuscript pretends to review the therapeutic possibilities of targeting the kynurenine pathway.

We concur with the Reviewer. The objective of this review was to identify potential therapeutic targets for diseases that are challenging to treat or deemed incurable.

In fact, the reader finds only a few isolated results, and they usually involve experimental studies on animal models. Whereas, IDO/TDO inhibitors have been subjected to multiple clinical trials, but these results have not been cited and systematized. Instead, a lot of space has been devoted for example to the herbal water extract (danshensu), whose composition is not defined and whose action is far from selective.

We meticulously searched the ClinicalTrials.gov database for clinical trials involving IDO inhibitors and identified 44 trials focusing on these inhibitors for cancer treatment. Additionally, one study explored a TDO inhibitor in patients with advanced solid tumors. On July 31, 2024, research will commence on diclofenac as a KMO inhibitor in individuals with alcohol use disorders. Consequently, no clinical studies are verifying the efficacy of selective IDO, TDO, or KMO inhibitors in treating the diseases discussed in this paper.

Given the inadequacy of effective pharmacotherapy, alternative treatment options are being actively pursued and evaluated. One such avenue is a return to natural medicine, which is why danshensu is mentioned in this manuscript as an ingredient in some traditional Chinese medicines.

The accuracy and reliability of the data in the manuscript is highly questionable. An example is the effect of kynurenine pathway metabolites on skin wound healing. Publication on the effect of kynurenine on wound healing (DOI: 10.1371/journal.pone.0091955 ) was omitted, particularly since clinical trials based on these results were conducted in humans (DOI: 10.36849/jdd.6197). There are more important publications on the subject (DOI: 10.1089/wound.2023.0137; DOI: 10.1007/s11010- 017-2975-3 ). Similarly, there is an extensive literature on inflammatory gastrointestinal diseases that has not been cited.

Thank you for this creative criticism. We believe it has enabled us to improve this manuscript. Following the reviewer's suggestion, we cited the four mentioned works (DOI: 10.1371/journal.pone.0091955; DOI: 10.36849/jdd.6197;DOI: 10.1089/wound.2023.0137; DOI: 10.1007/s11010-017-2975-3), as also we have expanded section 4.2. Pharmacological inhibition of KMO activity in the subsection Intestinal disorders. Additionally, we have completed section 4.1. Pharmacological modulation activity of IDO activity / 4.1.1. Inhibition of the enzyme / Intestinal disorders with four additional citations. When selecting literature for the manuscript, we were guided primarily by studies emphasizing the impact of enzyme modulation on selected diseases. This meant that some of the studies seemed to have been omitted by us. However, our goal was never to modulate the reader.

In conclusion, a review paper does not benefit the reader, and may even lead to an incorrect interpretation of the phenomena outlined in it. The final conclusion about the need for further research is trivial.

The manuscript has been revised following the suggestions of three reviewers. We acknowledge that the figures could have been more precise and that the text did not include all relevant published research results. Based on the reviewers' recommendations, all figures have been corrected, and the suggested publications have been incorporated into the text. Additionally, we have included references to new, recently published literature. Furthermore, we have revised the final conclusion.

Text chance: This underscores the importance of further research on this topic, particularly concerning their clinical application.

Reviewer 3 Report

Comments and Suggestions for Authors

Mor A et al's submission (Kynurenines As A Novel Target For The Treatment Of Inflammatory Disorders) is a comprehensive discussion of the kynurenine pathway in autoimmune, non-autoimmune, infectious and other disorders.  The scope and depth of the review is good and the writing is of a scholarly standard.

There are a few minor issues.

IDO1/IDO2  
In the Abstract (Ln 15) and in other places the authors give the impression that IDO1 and IDO2 are somewhat interchangeable.
E.g. at line 149 they refer to the "activity of the IDO1 isoform".  IDO1 and IDO2 are separate genes which are translated to different (but similar) proteins.
>Please ensure this is made more clear.

Ln 53 "It has been proved..." Nothing is proven, it remains undisproved.  Suggested rewording "There is strong evidence demonstrating that changes in the..."

Ln 63 TNF[alpha] is still in common use although there is a trend towards using just TNF as TNF[beta] has fallen out of use.

Ln 74 TH1 - Th1

Figure 1 "Limphocyte T" "Limphocyte B"
Ln 82, 83 "interleukine"

Ln 142 Please provide a citation for N-formylkynurenine being a ligand of AhR.

Ln 120, 220, 366, 480 "(Figure 3)" -  Is "(Figure 3)" meant to be part of the title?

Author Response

We would like to take this opportunity to deeply thank the Reviewer who identified the parts of our manuscript that required corrections or modifications. Please find the response to the Rewiever’s comments below.

Reviewer #3:

IDO1/IDO2  
In the Abstract (Ln 15) and in other places the authors give the impression that IDO1 and IDO2 are somewhat interchangeable. In the revised manuscript, we have clarified the differences and specific roles of IDO1 and IDO2 more precisely. Here are the specific changes made:
E.g. at line 149 they refer to the "activity of the IDO1 isoform".  IDO1 and IDO2 are separate genes which are translated to different (but similar) proteins.
>Please ensure this is made more clear.

Thank you for your valuable feedback. We apologize for any confusion regarding the interchangeability of IDO1 and IDO2. Our intent was not to imply that these enzymes are interchangeable, but rather to highlight their distinct yet sometimes overlapping roles in biological processes. In the revised manuscript, we have clarified the differences and specific roles of IDO1 and IDO2 more precisely. Here are the specific changes made:

Text changes:

Line 14-17: Key KP enzymes such as indoleamine 2,3-dioxygenase 1 (IDO1), indoleamine 2,3-dioxygenase 2 (IDO2), tryptophan 2,3-dioxygenase (TDO), and kynurenine 3-monooxygenase (KMO) have been considered promising therapeutic targets.

Throughout the manuscript: We have carefully reviewed the text to ensure that whenever IDO1 and IDO2 are mentioned, their distinct roles and properties are delineated. For instance:

Line 36-37: IDO1 is observed in almost all body tissues, while TDO activity is highest in the liver [10].

Lines 136-139: Thus, increased synthesis of KYN, mainly by IDO1, suppresses the immune system response, causing inactivation and apoptosis of TH1 and effector T cells, as well as activation of immunosuppressive T regulatory cells (Tregs) [28].

Lines 156-159: This indicates that IDO1 expression can also be elevated by autocrine stimulation in inflammatory conditions [29–31]. Conversely, a decrease in IL-6 or STAT3 expression reduces IDO1 activity and KYN synthesis [31,32], while blocking AhR signaling restores T cell proliferation and activation [33].

Lines 151: IDO1 - indoleamine 2,3-dioxygenase 1

Line 196: 3.1. Pharmacological Inhibition of enzyme of IDO1 and TDO activity

Line 276: 3.2. Pharmacological Stimulation of IDO1 Activity in Autoimmune Diabetes

Line 297: 4.1. Pharmacological modulation activity of IDO1 activity

Line 515: 5.1. Pharmacological Inhibition of IDO1 Activity

Ln 53 "It has been proved..." Nothing is proven, it remains undisproved.  Suggested rewording "There is strong evidence demonstrating that changes in the..."

Thank you for your valuable attention. We apologize for the imprecise language and have revised the sentence to better reflect the current state of the evidence. Here is the corrected text:

Text change: There is strong evidence demonstrating that in the changes in the activity of KP enzymes are observed in the development of a wide range of systemic disorders [11], including autoimmune [12], infectious [13], and metabolic diseases [14].

Ln 63 TNF[alpha] is still in common use although there is a trend towards using just TNF as TNF[beta] has fallen out of use.

Thank you for your attention. As suggested by the reviewer, we changed the notation of TNFa to TNF.

Lines 85-87: The activation of KP is induced mainly by pro-inflammatory factors such as interleukins (IL-1 and IL-6), tumor necrosis factor (TNF), and interferon-γ (IFN-γ), and it is observed during inflammation [21].

Line 153: TNF - tumor necrosis factor

Line 180: TNF - tumor necrosis factor

Lines 499-502: These results were confirmed by studies conducted in patients with active UC, in which both KMO and KYNU expression were positively correlated with the inflammatory factors TNF and IL-1β.

Ln 74 TH1 - Th1

Thank you for pointing out this incorrect entry. We have corrected it as suggested.

Lines 112-114: Thus, increased synthesis of KYN, mainly by IDO1, suppresses the immune system response, causing inactivation and apoptosis of Th1 and effector T cells, as well as activation of immunosuppressive T regulatory cells (Tregs) [28].

Figure 1 "Limphocyte T" "Limphocyte B"

Thank you for bringing this to our attention. We apologize for the typographical errors in Figure 1. The labels "Limphocyte T" and "Limphocyte B" have been corrected to "Lymphocyte T" and "Lymphocyte B," respectively.

Ln 82, 83 "interleukine"

Thank you for identifying the typographical error. We have corrected "interleukine" to "interleukin" in lines 126 and 127.

Line 154 (previously lines 126-127): IL-6 - interleukin 6, IL-1 - interleukin 1, IL-4 - interleukin 4, IL-10 - interleukin 10

Ln 142 Please provide a citation for N-formylkynurenine being a ligand of AhR.

Thank you for pointing out the error regarding N-formylkynurenine. The text should indeed refer to kynurenine as a ligand of the aryl hydrocarbon receptor (AhR). Here is the revised text along with the appropriate citation: Shadboorestan, A.; Koual, M.; Dairou, J.; Coumoul, X. The Role of the Kynurenine/AhR Pathway in Diseases Related to Metabolism and Cancer. Int. J. Tryptophan Res. IJTR 2023, 16, 11786469231185102, doi:10.1177/11786469231185102.

Text change: KYN is a compound with potent immunosuppressive properties and is an agonist of the aryl hydrocarbon receptor (AhR), which plays an important role in regulating numerous cellular signaling pathways and maintaining cellular homeostasis. Another AhR agonist is KYNA [25].

Ln 120, 220, 366, 480 "(Figure 3)" -  Is "(Figure 3)" meant to be part of the title?

Thank you for your observation. The references to "(Figure 3)" were not intended to be part of the titles. We have corrected these instances to ensure clarity.

Here are the revised sections:

Line 120: Removed "(Figure 3)" from the title and placed it appropriately within the text.

Line 220: Removed "(Figure 3)" from the title and placed it appropriately within the text.

Line 366: Removed "(Figure 3)" from the title and placed it appropriately within the text.

Line 480: Removed "(Figure 3)" from the title and placed it appropriately within the text.

We appreciate your attention to detail and believe this adjustment will improve the readability and accuracy of the manuscript.

Round 2

Reviewer 2 Report

Comments and Suggestions for Authors

I have no comments for Authors.